# Neural Discourse Structure for Text Categorization

## Abstract

We show that discourse structure, as defined by Rhetorical Structure Theory and provided by an existing discourse parser, benefits text categorization. Our approach uses a recursive neural network and a newly proposed attention mechanism to compute a representation of the text that focuses on salient content, from the perspective of both RST and the task. Experiments consider variants of the approach and illustrate its strengths and weaknesses.

## 1 Introduction

Advances in text categorization have the potential to improve systems for analyzing sentiment, inferring authorship or author attributes, making predictions, and many more. Several past researchers have noticed that methods that reason about the relative saliency or importance of passages within a text can lead to improvements (Ko et al., 2004). Latent variables (Yessenalina et al., 2010), structured-sparse regularizers (Yogatama and Smith, 2014), and neural attention models (Yang et al., 2016) have all been explored.

**Discourse structure**, which represents the organization of a text as a tree (for an example, see Figure 1), might provide cues for the importance of different parts of a text. Some promising results on sentiment classification tasks support this idea: Bhatia et al. (2015) and Hogenboom et al. (2015) applied hand-crafted weighting schemes to the sentences in a document, based on its discourse structure, and showed benefit to sentiment polarity classification.

In this paper, we investigate the value of discourse structure for text categorization more broadly, considering five tasks, through the use of a recursive neural network built on an

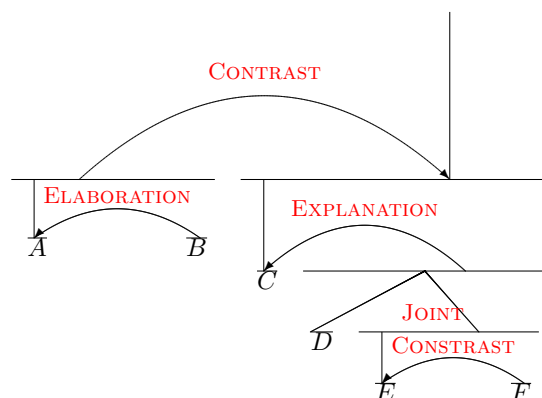

[Although the food was amazing]$^A$ [and I was in love with the spicy pork burrito,]$^B$ [the service was really awful.]$^C$ [We watched our waiter serve himself many drinks.]$^D$ [He kept running into the bathroom]$^E$ [instead of grabbing our bill.]$^F$

Figure 1: A manually constructed example of the RST (Mann and Thompson, 1988) discourse structure on a text.

automatically-derived document parse from a top-performing, open-source discourse parser, DPLP (Ji and Eisenstein, 2014). Our models learn to weight the importance of a document's sentences, based on their positions and relations in the discourse tree. We introduce a new, unnormalized attention mechanism to this end.

Experimental results show that variants of our model outperform prior work on four out of five tasks considered. Our method unsurprisingly underperforms on the fifth task, making predictions about legislative bills—a rather different genre in which discourse conventions are quite different from those in the discourse parser's training data. Further experiments show the effect of discourse parse quality on text categorization performance, suggesting that future improvements to discourse parsing will pay off for text categorization, and validate our new attention mechanism.

Our implementation is available at `http://anonymous.link`.

## 2 Background: Rhetorical Structure Theory

Rhetorical Structure Theory (RST; Mann and Thompson, 1988) is a theory of discourse that has enjoyed popularity in NLP. RST posits that a document can be represented by a tree whose leaves are elementary discourse units (EDUs, typically clauses or sentences). Internal nodes in the tree correspond to spans of sentences that are connected via discourse relations such as CONTRAST and ELABORATION. In most cases, a discourse relation links adjacent spans denoted "nucleus" and "satellite," with the former more essential to the writer's purpose than the latter.[1]

An example of a manually constructed RST parse for a restaurant review is shown in Figure 1. The six EDUs are indexed from $A$ to $F$; the discourse tree organizes them hierarchically into increasingly larger spans, with the last CONTRAST relation resulting in a span that covers the whole review. Within each relation, the RST tree indicates the nucleus pointed by an arrow from its satellite (e.g., in the ELABORATION relation, $A$ is the nucleus and $B$ is the satellite).

The information embedded in RST trees has motivated many applications in NLP research, including document summarization (Marcu, 1999), argumentation mining (Azar, 1999), and sentiment analysis (Bhatia et al., 2015). In most applications, RST trees are built by automatic discourse parsing, due to the expensive cost of manual annotation. In this work, we use a state-of-the-art open-source RST-style discourse parser, DPLP (Ji and Eisenstein, 2014).[2]

We follow recent work that suggests transforming the RST tree into a dependency structure (Yoshida et al., 2014).[3] Figure 2(a) shows the corresponding dependency structure of the RST tree in Figure 1. It is clear that $C$ is the root of the tree, and in fact this clause summarizes the review and suffices to categorize it as negative. This dependency representation of the RST tree offers a

---

[1]There are also a few exceptions in which a relation can be realized with multiple nuclei.

[2]`https://github.com/jiyfeng/DPLP`

[3]The transformation is trivial and deterministic given the nucleus-satellite mapping for each relation. The procedure is analogous to the transformation of a headed phrase-structure parse in syntax into a dependency tree (e.g., Yamada and Matsumoto, 2003).

form of inductive bias for our neural model, helping it to discern the most salient parts of a text in order to assign it a label.

## 3 Model

Our model is a recursive neural network built on a discourse dependency tree. It includes a distributed representation computed for each EDU, and a composition function that combines EDUs and partial trees into larger trees. At the top of the tree, the representation of the complete document is used to make a categorization decision. Our approach is analogous to (and inspired by) the use of recursive neural networks on *syntactic* dependency trees, with word embeddings at the leaves (Socher et al., 2014).

### 3.1 Representation of Sentences

Let $\mathbf{e}$ be the distributed representation of an EDU. We use a bidirectional LSTM on the words' embeddings within each EDU (details of word embeddings are given in section 4), concatenating the last hidden state vector from the forward LSTM ($\overleftarrow{\mathbf{e}}$) with that of the backward LSTM ($\overrightarrow{\mathbf{e}}$) to get $\mathbf{e}$.

There is extensive recent work on architectures for embedding representations of sentences and other short pieces of text, including, for example, (bi)recursive neural networks (Paulus et al., 2014) and convolutional neural networks (Kalchbrenner et al., 2014). Future work might consider alternatives; we chose the bidirectional LSTM due to its effectiveness in many settings.

### 3.2 Full Recursive Model

Given the discourse dependency tree for an input text, our recursive model builds a vector representation through composition at each arc in the tree. Let $\mathbf{v}_i$ denote the vector representation of EDU $i$ and its descendants. For the base case where EDU $i$ is a leaf in the tree, we let $\mathbf{v}_i = \tanh(\mathbf{e}_i)$, which is the elementwise hyperbolic tangent function.

For an internal node $i$, the composition function considers a parent and all of its children, whose indices are denoted by $children(i)$. In defining this composition function, we seek for (i.) the contribution of the parent node $\mathbf{e}_i$ to be central; and (ii.) the contribution of each child node $\mathbf{e}_j$ be determined by its content as well as the discourse relation it holds with the parent. We therefore define

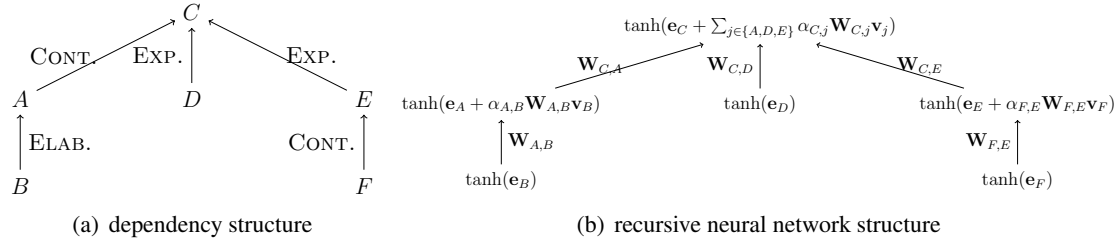

(a) dependency structure (b) recursive neural network structure

Figure 2: (a.) The discourse dependency tree derived from the example RST tree in Figure 1. (b.) The corresponding recursive neural network model built on the tree.

$$\mathbf{v}_i = \tanh\left(\mathbf{e}_i + \sum_{j \in children(i)} \alpha_{i,j} \mathbf{W}_{r_{i,j}} \mathbf{v}_j\right),\tag{1}$$

where $\mathbf{W}_{r_{i,j}}$ is a relation-specific composition matrix indexed by the relation between $i$ and $j$, $r_{i,j}$.

$\alpha_{i,j}$ is an "attention" weight, defined as

$$\alpha_{i,j} = \sigma\left(\mathbf{e}_i^\top \mathbf{W}_\alpha \mathbf{v}_j\right),\tag{2}$$

where $\sigma$ is the elementwise sigmoid and $\mathbf{W}_\alpha$ contains attention parameters (these are relation-independent). Our attention mechanism differs from prior work (Bahdanau et al., 2015), in which attention weights are normalized to sum to one across competing candidates for attention. Here, $\alpha_{i,j}$ does not depend on node $i$'s other children. This is motivated by RST, in which the presence of a node does not signify lesser importance to its siblings. Consider, for example, EDU $D$ and text span $E$-$F$ in Figure 1, which in parallel provide EXPLANATION for EDU $C$. This scenario differs from machine translation, where attention is used to implicitly and softly align output-language words to relatively few input-language words. It also differs from attention in composition functions used in syntactic parsing (Kuncoro et al., 2017), where attention can mimic head rules that follow from an endocentricity hypothesis of syntactic phrase representation.

Our recursive composition function, through the attention mechanism and the relation-specific weight matrices, is designed to learn how to differently weight EDUs for the categorization task. This idea of using a weighting scheme along with discourse structure is explored in prior works (Bhatia et al., 2015; Hogenboom et al., 2015), although they are manually designed, rather than learned from training data.

Once we have $\mathbf{v}_{root}$ of a text, the prediction of its category is given by softmax $(\mathbf{W}_o \mathbf{v}_{root} + \mathbf{b})$.

We refer to this model as the FULL model, since it makes use of the entire discourse dependency tree.

### 3.3 Unlabeled Model

The FULL model based on Equation 1 uses a dependency discourse tree with relations. Because alternate discourse relation labels have been proposed (e.g., Prasad et al., 2008), we seek to measure the effect of these labels. We therefore consider an UNLABELED model based only on the tree structure, without the relations:

$$\mathbf{v}_i = \tanh\left(\mathbf{e}_i + \sum_{j \in children(i)} \alpha_{i,j} \mathbf{v}_j\right).\tag{3}$$

Here, only attention weights are used to compose the children nodes' representations, significantly reducing the number of model parameters.

This UNLABELED model is similar to the depth weighting scheme introduced by Bhatia et al. (2015), which also uses an unlabeled discourse dependency tree, but our attention weights are computed by a function whose parameters are learned. This approach sits squarely between Bhatia et al. (2015) and the flat document structure used by Yang et al. (2016); the UNLABELED model still uses discourse to bias the model toward some content (that which is closer to the tree's root).

### 3.4 Simpler Variants

We consider two additional baselines that are even simpler. The first, ROOT, uses the discourse dependency structure only to select the root EDU, which is used to represent the entire text: $\mathbf{v}_{root} = \mathbf{e}_{root}$. No composition function is needed. This model variant is motivated by work on document summarization (Yoshida et al., 2014), where the

most central EDU is used to represent the whole text.

The second variant, ADDITIVE, uses all the EDUs with a simple composition function, and does not depend on discourse structure at all: $\mathbf{v}_{root} = \frac{1}{N} \sum_{i=1}^{N} \mathbf{e}_i$, where $N$ is the total number of EDUs. This serves as a baseline to test the benefits of discourse, controlling for other design decisions and implementation choices. Although sentence representations $\{\mathbf{e}_i\}$ are built in a different way from the work of Yang et al. (2016), this model is quite similar to their HN-AVE model on building document representations.

## 4 Implementation Details

The parameters of all components of our model (top-level classification, composition, and EDU representation) are learned end-to-end using standard methods. We implement our learning procedure with the `DyNet` package (Neubig et al., 2017).

**Preprocessing.** For all datasets, we use the same preprocessing steps, mostly following recent work from language modeling (e.g., Mikolov et al., 2010). We lowercased all the tokens and removed tokens that contain only punctuation symbols. We replaced numbers in the documents with a special number token. Low-frequency word types were replaced by UNK; we reduce the vocabulary for each dataset until approximately 5% of tokens are mapped to UNK. The vocabulary sizes after preprocessing are also shown in Table 1.

**Discourse parsing.** Our model requires the discourse structure for each document. We used DPLP, the RST parser from Ji and Eisenstein (2014), which is one of the best discourse parsers on the RST discourse treebank benchmark (Carlson et al., 2001). It employs a greedy decoding algorithm for parsing, producing 2,000 parses per minute on average on a single CPU. DPLP provides discourse segmentation, breaking a text into EDUs, typically clauses or sentences, based on syntactic parses provided by Stanford CoreNLP. RST trees are converted to dependencies following the method of Yoshida et al. (2014). DPLP as distributed is trained on 347 Wall Street Journal articles from the Penn Treebank (Marcus et al., 1993).

**Word embeddings.** In cases where there are 10,000 or fewer training examples, we used pretrained GloVe word embeddings (Pennington et al., 2014), following previous work on neural discourse processing (Ji and Eisenstein, 2015). For larger datasets, we randomly initialize word embeddings and train them alongside other model parameters.

**Learning and hyperparameters.** Online learning was performed with the optimization method and initial learning rate as hyperparameters. To avoid the exploding gradient problem, we used the norm clipping trick with a threshold of $\tau = 5.0$. In addition, dropout rate 0.3 was used on both input and hidden layers to avoid overfitting. We performed grid search over the word vector representation dimensionality, the LSTM hidden state dimensionality (both $\{32, 48, 64, 128, 256\}$), the initial learning rate ($\{0.1, 0.01, 0.001\}$), and the update method (SGD and Adam, Kingma and Ba, 2015). For each corpus, the highest-accuracy combination of these hyperparameters is selected using development data or ten-fold cross validation, which will be specified in section 5.

## 5 Datasets

We selected five datasets of different sizes and corresponding to varying categorization tasks. Some information about these datasets are summarized in Table 1.

**Sentiment analysis on Yelp reviews.** Originally from the Yelp Dataset Challenge in 2015, this dataset contains 1.5 million examples. We used the preprocessed dataset from Zhang et al. (2015), which has 650,000 training and 50,000 test examples. The task is to predict an ordinal rating (1–5) from the text of the review. To select the best combination of hyperparameters, we randomly sampled 10% training examples as the development data. We compared with hierarchical attention networks (Yang et al., 2016), which use the normalized attention mechanism on both word and sentence layers with a flat document structure, and provide the state-of-the-art result on this corpus.

**Framing dimensions in news articles.** The Media Frames Corpus (MFC; Card et al., 2015) includes around 4,200 news articles about immigration from 13 U.S. newspapers over the years 1980–2012. The annotations of these articles are in terms of a set of 15 general-purpose labels, such as ECONOMICS and MORALITY, designed to categorize the emphasis framing applied to the

| Dataset | Task | Classes | Number of docs. | | | | Vocab. size |
| --- | --- | --- | --- | --- | --- | --- | --- |
| | | | Total | Training | Development | Test | |
| Yelp | Sentiment | 5 | 700K | 650K | – | 50K | 10K |
| MFC | Frames | 15 | 4.2K | – | – | – | 7.5K |
| Debates | Vote | 2 | 1.6K | 1,135 | 105 | 403 | 5K |
| Movies | Sentiment | 2 | 2.0K | – | – | – | 5K |
| Bills | Survival | 2 | 52K | 46K | – | 6K | 10K |

Table 1: Information about the five datasets used in our experiments. To compare with prior work, we use different experimental settings. For Yelp and Bill corpora, we use 10% of the training examples as development data. For MFC and Movies corpora, we use 10-fold cross validation and report averages across all folds.

immigration issue within the articles. We focused on predicting the single *primary* frame of each article. The state-of-the-art result on this corpus is from Card et al. (2016), where they used logistic regression together with unigrams, bigrams and Bamman-style personas (Bamman et al., 2014) as features. The best feature combination in their model alongside other hyperparameters was identified by a Bayesian optimization method (Bergstra et al., 2015). To select hyperparameters, we used a small set of examples from the corpus as a development set. Then, we report average accuracy across 10-fold cross validation as in (Card et al., 2016).

**Congressional floor debates.** The corpus was originally collected by Thomas et al. (2006), and the data split we used was constructed by Yessenalina et al. (2010). The goal is to predict the vote ("yea" or "nay") for the speaker of each speech segment. The most recent work on this corpus is from Yogatama and Smith (2014), which proposed structured regularization methods based on linguistic components, e.g., sentences, topics, and syntactic parses. Each regularization method induces a linguistic bias to improve text classification accuracy, where the best result we repeated here is from the model with sentence regularizers.

**Movie reviews.** This classic movie review corpus was constructed by Pang and Lee (2004) and includes 1,000 positive and 1,000 negative reviews. On this corpus, we used the standard ten-fold data split for cross validation and reported the average accuracy across folds. We compared with the work from both Bhatia et al. (2015) and Hogenboom et al. (2015), which are two recent works on discourse for sentiment analysis. Bha-

tia et al. (2015) used a hand-crafted weighting scheme to bias the bag-of-word representations on sentences. Hogenboom et al. (2015) also considered manually-designed weighting schemes and a lexicon-based model as classifier, achieving performance inferior to fully-supervised methods like Bhatia et al. (2015) and ours.

**Congressional bill corpus.** This corpus, collected by Yano et al. (2012), includes 51,762 legislative bills from the 103rd to 111th U.S. Congresses. The task is to predict whether a bill will survive based on its content. We randomly sampled 10% training examples as development data to search for the best hyperparameters. To our knowledge, the best published results are due to Yogatama and Smith (2014), which is the same baseline as for the congressional floor debates corpus.

# 6 Experiments

We evaluated all variants of our model on the five datasets presented in section 5, comparing in each case to the published state of the art as well as the most relevant works.

**Results.** See Table 2. On four out of five datasets, our UNLABELED model (line 8) outperforms past methods. In the case of the very large Yelp dataset, our FULL model (line 9) gives even stronger performance, but not elsewhere, suggesting that it is overparameterized for the smaller datasets. Indeed, on the MFC and Movies tasks, the discourse-ignorant ADDITIVE outperforms the FULL model. On these datasets, the selected FULL model had nearly 20 times as many parameters as the UNLABELED model, which in turn had twice as many parameters as the ADDITIVE.

| Method | Yelp | MFC | Debates | Movies | Bills |
|---|---|---|---|---|---|
| *Prior work* | | | | | |
| 1. Yang et al. (2016) | 71.0 | — | — | — | — |
| 2. Card et al. (2016) | — | 56.8 | — | — | — |
| 3. Yogatama and Smith (2014) | — | — | 74.0 | — | 88.5 |
| 4. Bhatia et al. (2015) | — | — | — | 82.9 | — |
| 5. Hogenboom et al. (2015) | — | — | — | 71.9 | — |
| *Variants of our model* | | | | | |
| 6. ADDITIVE | 68.5 | **57.6** | 69.0 | 82.7 | 80.1 |
| 7. ROOT | 54.3 | 51.2 | 60.3 | 68.7 | 70.5 |
| 8. UNLABELED | **71.3** | **58.4** | **75.7** | **83.1** | 78.4 |
| 9. FULL | **71.8** | 56.3 | **74.2** | 79.5 | 77.0 |

Table 2: Test-set accuracy across five datasets. Results from prior work are reprinted from the corresponding publications. Boldface marks performance stronger than the previous state of the art.

This finding demonstrates the benefit of explicit discourse structure—even the output from an imperfect parser—for text categorization in some genres. Even though the discourse parser is trained on news text, it still offers benefit to restaurant and movie reviews and to the genre of congressional debates. Even for news text, if the training dataset is small (e.g., MFC), a lighter-weight variant of discourse (UNLABELED) is preferred.

Legislative bills, which have technical legal content and highly specialized conventions (see Appendix A in the supplementary material for an example), are arguably the most distant genre from news among those we considered. On that task, we see discourse working against accuracy. Note that the corpus of bills is more than ten times larger than three cases where our UNLABELED model outperformed past methods, suggesting that the drop in performance is not due to lack of data.

It is also important to notice that the ROOT model performs quite poorly in all cases. This implies that discourse structure is not simply helping by finding a single EDU upon which to make the categorization decision.

**Qualitative analysis.** Figure 3 shows some example texts from the Yelp Review corpus with their discourse structures produced by DPLP, where the weights were generated with the FULL model. Figure 3(a) and 3(b) are two successful examples of the FULL model. Figure 3(a) shows a simple case with respect to the discourse structure. Figure 3(b) is slightly different—the text in this example may have more than one reasonable discourse structure, e.g., $2D$ could be a child of

$2C$ instead of $2A$. In both cases, discourse structures help the FULL model bias to the important sentences.

Figure 3(c), on the other hand, presents a negative example, where DPLP failed to identify the most salient sentence $3F$. In addition, the weights produced by the FULL model do not make much sense, which we suspect the model was confused by the structure. Figure 3(c) also presents a manually-constructed discourse structure on the same text for reference.

**Effect of parsing performance.** A natural question is whether further improvements to RST discourse parsing would lead to even greater gains in text categorization. While advances in discourse parsing are beyond the scope of this paper, we can gain some insight by exploring degradation to the DPLP parser. An easy way to do this is to train it on subsets of the RST discourse treebank. We repeated the conditions described above for our FULL model, training DPLP on 25%, 50%, and 75% of the training set (randomly selected in each case) before re-parsing the data for the sentiment analysis task. We did not repeat the hyperparameter search. In Figure 4, we plot accuracy of the classifier ($y$-axis) against the $F_1$ performance of the discourse parser ($x$-axis). Unsurprisingly, lower parsing performance implies lower classification accuracy. Notably, if the RST discourse treebank were reduced to 25% of its size, our method would underperform the discourse-ignorant model of Yang et al. (2016). While we cannot extrapolate with certainty, these findings suggest that further improvements to discourse

From DPLP:

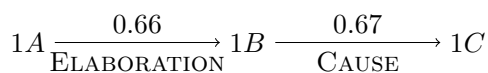

[This store is somewhat convenient but I can never find any workers,]$^{1A}$ [it drives me crazy.]$^{1B}$ [I never come here on the weekends or around holidays anymore.]$^{1C}$

(a) true label: 2, predicted label: 2

From DPLP:

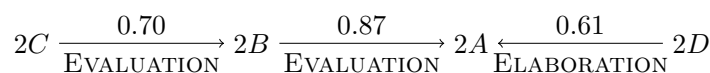

[I love these people.]$^{2A}$ [They are very friendly and always ask about my life.]$^{2B}$ [They remember things I tell them then ask about it the next time I'm in.]$^{2C}$ [Patrick and Lily are the best but everyone there is wonderful in their own ways.]$^{2D}$

(b) true label: 5, predicted label: 5

From DPLP:

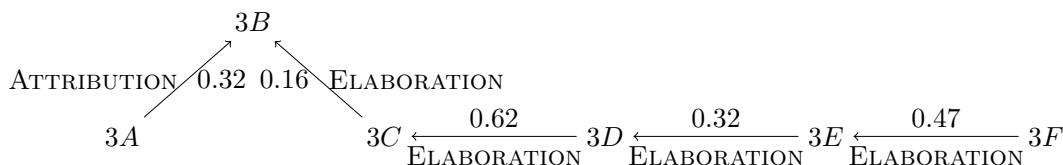

Manually constructed:

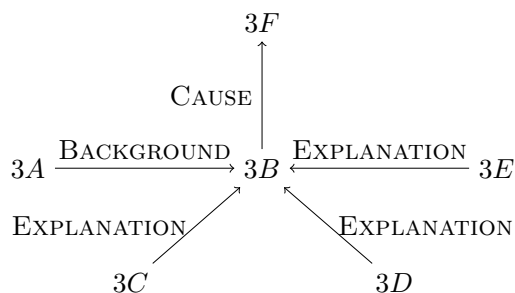

[We use to visit this pub 10 years ago because they had a nice english waitress and excellent fish and chips for the price.]$^{3A}$ [However we went back a few weeks ago and were disappointed.]$^{3B}$ [The price of the fish and chip dinner went up and they cut the portion in half.]$^{3C}$ [No one assisted us in putting two tables together we had to do it ourselves.]$^{3D}$ [Two guests wanted a good English hot tea and they didn't brew it in advance.]$^{3E}$ [So we've decided there are newer and better places to eat fish and chips especially up in north phoenix.]$^{3F}$

(c) true label: 1, predicted label: 3

Figure 3: Some example texts (with light revision for readability) from the Yelp Review corpus and their corresponding dependency discourse parses. The numbers on dependency edges are attention weights produced by the FULL model.

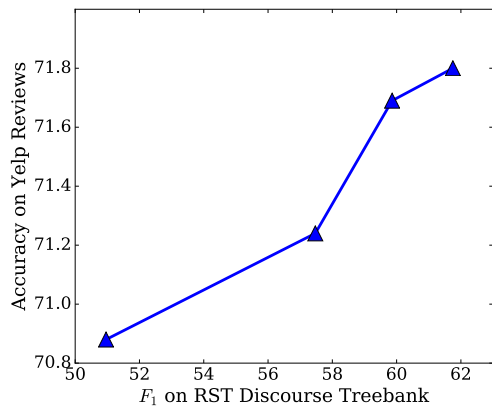

Figure 4: Varying the amount of training data for the discourse parser, we can see how parsing $F_1$ performance affects accuracy on the Yelp review task.

parsing, through larger annotated datasets or improved models, could lead to greater gains.

**Attention mechanism.** In section 3, we contrasted our new attention mechanism (Equation 2), which is inspired by RST's lack of "competition" for salience among satellites, with the attention mechanism used in machine translation (Bahdanau et al., 2015). We consider here a variant of our model with normalized attention:

$$\boldsymbol{\alpha}'_i = \mathrm{softmax}\left(\begin{bmatrix} \vdots \\ \mathbf{v}_j^\top \\ \vdots \end{bmatrix}_{j \in children(i)} \mathbf{W}_\alpha \cdot \mathbf{e}_i\right).$$
$$(4)$$

The result here is a vector $\boldsymbol{\alpha}'_i$, with one element for each child node $j \in children(i)$, and which sums to one.

This variant of the FULL model achieves 70.3% accuracy, giving empirical support to our theoretically-motivated design decision not to normalize attention. Of course, further architecture improvements may yet be possible.

## 7    Related Work

Early work on text categorization often treated text as a bag of words (e.g., Joachims, 1998; Yang and Pedersen, 1997). Representation learning, for example through matrix decomposition (Deerwester et al., 1990) or latent topic variables (Ramage et al., 2009), has been considered to avoid overfitting in the face of sparse data.

The assumption that all parts of a text should influence categorization equally persists even as

more powerful representation learners are considered. Zhang et al. (2015) treat a text as a sequence of characters, proposing to a deep convolutional neural network to build text representation. Xiao and Cho (2016) extended that architecture by inserting a recurrent neural network layer between the convolutional layer and the classification layer.

In contrast, our contributions follow Ko et al. (2004), who sought to weight the influence of different parts of an input text on the task. Two works that sought to learn the importance of sentences in a document are Yessenalina et al. (2010) and Yang et al. (2016). The former used a latent variable for the informativeness of each sentence, and the latter used a neural network to learn an attention function. Neither used any linguistic bias, relying only on task supervision to discover the latent variable distribution or attention function. Our work builds the neural network directly on a discourse dependency tree, favoring the most central EDUs over the others but giving the model the ability to overcome this bias.

Another way to use linguistic information was presented by Yogatama and Smith (2014), who used a bag-of-words model. The novelty in their approach was a data-driven regularization method that encouraged the model to collectively ignore groups of features found to coocur. Most related to our work is their "sentence regularizer," which encouraged the model to try to ignore training-set sentences that were not informative for the task. Discourse structure was not considered.

**Discourse for sentiment analysis.** Recently, discourse structure has been considered for sentiment analysis, which can be cast as a text categorization problem. Bhatia et al. (2015) proposed two discourse-motivated models for sentiment polarity prediction. One of the models is also based on discourse dependency trees, but using a handcrafted weighting scheme. Our method's attention mechanism automates the weighting.

## 8    Conclusion

We conclude that automatically-derived discourse structure is often helpful to categorization of many kinds of text, and the benefit increases with the accuracy of discourse parsing. This effect reverses for legislative bills, a text genre whose discourse structure diverges from that of news. These findings motivate further improvements to discourse parsing, especially for new genres.

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
