# Peer review of "Neural Discourse Structure for Text Categorization"

_ACL 2017 — decision unknown_

[Official Review · Reviewer 1 · rating 4 · confidence 4]
soundness 5 · originality 5 · clarity 4 · impact 3 · substance 4 · appropriateness 5 · meaningful comparison 3 · presentation format Oral Presentation

This paper proposed to explore discourse structure, as defined by Rhetorical
Structure Theory (RST) to improve text categorization. A RNN with attention
mechanism is employed to compute a representation of text. The experiments on
various of dataset shows the effectiveness of the proposed method. Below are my
comments:

(1) From Table 2, it shows that “UNLABELED” model performs better on four
out of five datasets than the “FULL” model. The authors should explain more
about this, because intuitively, incorporating additional relation labels
should bring some benefits. Is the performance of relation labelling so bad and
it hurts the performance instead?

(2) The paper also transforms the RST tree into a dependency structure as a
pre-process step. Instead of transforming, how about keep the original tree
structure and train a hierarchical model on that?

(3) For the experimental datasets, instead of comparing with only one dataset
with each of the previous work, the authors may want to run experiments on more
common datasets used by previous work.

[Official Review · Reviewer 2 · rating 3 · confidence 3]
soundness 5 · originality 5 · clarity 5 · impact 3 · substance 4 · appropriateness 5 · meaningful comparison 3 · presentation format Poster

- Strengths:

The main strength of this paper is the incorporation of discourse structure in
the DNN's attention model, which allows the model to learn the weights given to
different EDUs.

Also the paper is very clear, and provides a good explanation of both RST and
how it is used in the model.
Finally, the evaluation experiments are conducted thoroughly with strong,
state-of-the-art baselines.

- Weaknesses:

The main weakness of the paper is that the results do not strongly support the
main claim that discourse structure can help text classification. Even the
UNLABELED variant, which performs best and does outperform the state of the
art, only provides minimal gains (and hurts in the legal/bills domain). The
approach (particularly the FULL variant) seems to be too data greedy but no
real solution is provided to address this beyond the simpler UNLABELED and ROOT
variants.

- General Discussion:

In general, this paper feels like a good first shot at incorporating discourse
structure into DNN-based classification, but does not fully convince that
RST-style structure will significantly boost performance on most tasks (given
that it is also very costly to build a RST parser for a new domain, as would be
needed in the legal/bill domains described in this paper). I wish the authors
had explored or at least mentioned next steps in making this approach work, in
particular in the face of data sparsity. For example, how about defining
(task-independent) discourse embeddings? Would it be possible to use a DNN for
discourse parsing that could be incorporated in the main task DNN and optimized
jointly  end-to-end? Again, this is good work, I just wish the authors had
pushed it a little further given the mixed results.